# One Token per Highly Selective Frame: Towards Extreme Compression for Long Video Understanding

**Zheyu Zhang**
University of Illinois Urbana-Champaign
`zheyuz5@illinois.edu`

**Ziqi Pang**
University of Illinois Urbana-Champaign
`ziqip2@illinois.edu`

**Shixing Chen**
Amazon Prime Video
`shixic@amazon.com`

**Xiang Hao**
Amazon Prime Video
`xianghao@amazon.com`

**Vimal Bhat**
Amazon Prime Video
`vimalb@amazon.com`

**Yu-Xiong Wang**
University of Illinois Urbana-Champaign
`yxw@illinois.edu`

## Abstract

Long video understanding is inherently challenging for vision-language models (VLMs) because of the extensive number of frames. With each video frame typically expanding into tens or hundreds of tokens, the limited context length of large language models (LLMs) forces the VLMs to perceive the frames sparsely and lose temporal information. To address this, we explore extreme video token compression towards *one token per frame* at the final LLM layer. Our key insight is that heuristic-based compression, widely adopted by previous methods, is prone to information loss, and this necessitates supervising LLM layers into *learnable* and *progressive* modules for *token-level compression* (LP-Comp). Such compression enables our VLM to digest 2x-4x more frames with improved performance. To further increase the token efficiency, we investigate *frame-level compression*, which selects the frames most relevant to the queries via the internal attention scores of the LLM layers, named *question-conditioned compression* (QC-Comp). As a notable distinction from previous studies, we mitigate the position bias of LLM attention in long contexts, *i.e.*, the over-concentration on the beginning and end of a sequence, by splitting long videos into short segments and employing local attention. Collectively, our combined *token-level* and *frame-level* leads to an e**x**treme **comp**ression model for long video understanding, named **XComp**, achieving a significantly larger compression ratio and enabling denser frame sampling. Our XComp is finetuned from VideoChat-Flash with a data-efficient *supervised compression tuning* stage that only requires 2.5% of the supervised fine-tuning data, yet boosts the accuracy from 42.9% to 46.2% on LVBench and enhances multiple other long video benchmarks. Code and Model are available at `https://github.com/ZheyuAqaZhang/XComp`.

## 1 Introduction

Enabling vision-language models (VLMs) to comprehensively understand long videos remains a critical yet formidable challenge [12, 22, 23, 32, 35, 38, 57, 61, 62, 69, 70]. The sheer volume of visual information challenges current VLMs' inherent context length constraints: as each frame is

39th Conference on Neural Information Processing Systems (NeurIPS 2025).

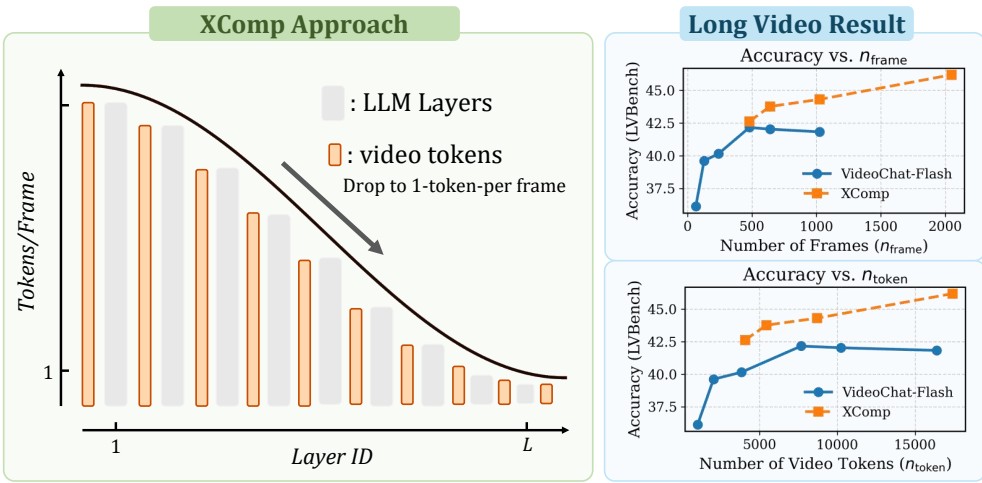

Figure 1: **Left:** We present XComp that explores using the LLM layers to progressively compress the video tokens towards the extreme of *one token per frame*. **Right:** Such capabilities enable the model to better improve itself with denser input frames without significantly increasing video tokens.

encoded into tens or hundreds of tokens, processing more than 1k frames usually exceeds the typical context lengths or computational budgets of the large language model (LLM) layers for the VLMs [11] during both training and inference. Yet, such a framework is still far from processing an hour-long video at more than one frame per second (FPS), easily losing critical temporal information and visual details. Therefore, how to effectively *compress the tokens of video frames while maintaining useful information* becomes the major bottleneck. In this paper, we explore the extreme of this direction and propose an approach to compress each video frame into *one single highly informative token* when reaching the final layer of the LLM in VLMs, thereby unlocking an unprecedented density of frames for VLMs to process.

From this aspect, we aim to achieve extreme compression at the *token level*: condensing the information from tens or hundreds of tokens per frame into a single token. Existing methods predominantly rely on heuristic-based or training-free strategies, such as special token selection [19, 35, 53] and pooling [31, 34]. Despite their reduced token counts, these methods risk discarding crucial visual tokens, as their underlying LLM layers of the VLMs were not supervised to encapsulate the capability of compression, *i.e.*, condense the contexts from the dropped tokens into the kept ones. As a result, our analysis (Table 2) reveals the limitations of such heuristic approaches when reaching high compression ratios. Therefore, our key insight is that the LLM layers should be supervised to compress extensive video data into remarkably fewer representative tokens via an additional *supervised compression tuning* stage, motivating *learnable* compression. To further avoid drastically losing information at large compression ratios, we let the LLM layers *progressively* compress the visual tokens with a smooth schedule. These two combined lead to our "*learnable* and *progressive* compression" (LP-Comp) as in Fig. 1, which marks clear distinctions with previous heuristic methods.

In addition to decreasing the number of tokens per frame, the token efficiency of long video understanding can be further enhanced by selecting the most relevant frames, corresponding to the compression at the *frame level*. Within the internal representation of VLMs, the distinctions of such relevant frames can be intuitively evaluated by the attention scores between the questions and the video tokens [19], where the frames having larger attention scores are preferred. However, a bottleneck under-explored by the previous methods is the internal *position bias* of LLM transformers in long context understanding: the attention heads might assign larger scores to the tokens at the beginning or the end of the sequence [42]. Inspired by local attention [3, 47, 76], under the context of long video understanding, this motivates a splitting of long videos into short video chunks for frame-level compression, where our model inspects the relevance of frames inside each video segment. Such a step effectively utilizes the question information and is referred to as *question-conditioned* compression (QC-Comp).

Combining both *learnable & progressive* compression (LP-Comp) and *question-conditioned* compression (QC-Comp), we propose a VLM named **XComp** that could achieve extreme token efficiency

towards *one-token-per-frame for highly selective frames*. Notably, the compression behavior of the LLM layers can be learned via a data-efficient *supervised compression tuning* stage requiring only 2.5% of the supervised fine-tuning (SFT) data of VideoChat-Flash [35]. More importantly, such token efficiency enables us to sample the frames densely and significantly enhance the performance as the LVBench [59] evaluation in Fig. 1: our method not only achieves better accuracy on a token-for-token efficiency basis, but also keeps improving with the number of input frames increases, while VideoChat-Flash experiences performance drops after a certain frame number.

In summary, we make the following contributions:

1. We introduce **LC-Comp**, a *learnable* and *progressive* framework that supervises LLM layers to compress at the token level, without purely relying on heuristics.
2. We propose **QC-Comp**, a frame-level *question-conditioned* selection method that effectively selects the relevant frames for specific questions.
3. We demonstrate the effectiveness of our token compression towards *each selected video frame as a single token*, drastically increasing the number of frames that can be processed and leading to improved long video understanding accuracy.

## 2   Related Work

**Vision-Language Models for Long Sequence Understanding.**   Early Vision-Language Models (VLMs), such as GPT-4V and Gemini-1.5 [49, 58], showcased powerful multimodal reasoning by integrating visual encoders with large language models. Open-source efforts like Llama-Vid [36], IDEFICS [24], VideoChat [34], Video-LLaMA [12], and others [2, 32, 35, 38, 44, 61, 62] have further advanced capabilities, often matching or exceeding proprietary systems on various benchmarks. While effective on static images or short video clips, scaling VLMs to long videos (minutes to hours) introduces a significant challenge. The core difficulty lies in managing the immense volume of visual data, which generates an excessive number of tokens that quickly exceed the typical context windows and computational capacities of the LLM components during both training and inference. Current research addressing long-context VLMs primarily explores two directions: (i) methods that significantly extend the effective context length of the underlying transformer architectures, such as LongVA, LongVILA, LongViTA, and LLaVA-NeXT [11, 55, 75, 77], and (ii) strategies focused on reducing the number of visual tokens fed to the LLM via selection or compression modules [26, 28, 56, 57]. While context extension allows processing more tokens, it often incurs high computational costs. Conversely, existing visual compression techniques, while reducing token count, frequently rely on heuristics or fixed operations, or uniform sampling across the video [2, 12, 32, 34, 35, 75] that struggle to preserve critical temporal and fine-grained information, particularly when pushed to high compression ratios necessary for very long videos. Other orthogonal approaches tackle long videos by processing shorter segments independently and employing retrieval mechanisms [3].

**Video Token Compression in Multimodal LLMs.**   Given the computational constraints of processing dense video streams with VLMs, compressing visual tokens is a vital approach. Various methods have been proposed to distill frame sequences into more manageable representations. These include temporal pooling strategies [17, 46], heuristic or saliency-based frame selection often guided by simple visual priors or attention mechanisms [56, 57], and lightweight learned modules or bottlenecks aimed at reducing the token sequence, such as Llama-Vid [36] which explores compressing frames potentially to a single token by leveraging cross-attention with a text query. Relatedly, techniques for dynamic token pruning or merging have been explored for both images and video [5, 13, 26, 29, 45, 52, 68, 70]. However, a critical limitation of many existing approaches, especially those relying purely on heuristics or fixed pooling, is that they are not trained to consolidate the information from dropped tokens into the retained ones within the LLM's representation space. Aggressively applying these methods to achieve the extreme compression ratios necessary for very long videos risks discarding vital visual details, thus hindering deep understanding. Furthermore, while frame selection is intuitive, applying it effectively across extremely long sequences without being hampered by issues like internal LLM position biases remains challenging. Therefore, developing methods that can perform learned and highly efficient visual compression, potentially towards representing each frame with an unprecedentedly low token count while preserving critical information, remains a significant research frontier.

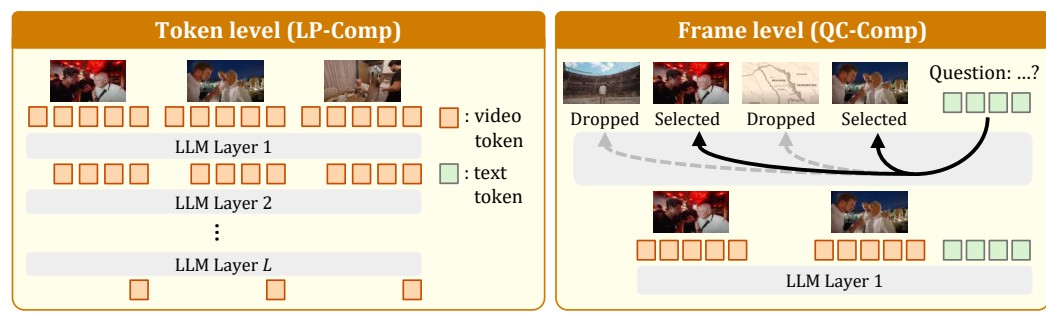

Figure 2: **Overview.** Our XComp comprises of two parts to achieve extreme compression in long video understanding. **(1)** At the *token level*, we propose the supervised compression tuning that enables the LLM to compress every video frame into one compact token in a *learnable and progressive* manner, namely, LP-Comp (Sec. 3.2). **(2)** At the *frame level*, we utilize the internal attention mechanism to select the frames relevant to the questions, which is *question-conditioned* compression, namely, QC-Comp (Sec. 3.3). Both aspects combined lead to our objective of extreme compression in long video understanding: one token per highly selective frame.

## 3 Methodology

Towards the objective of extreme compression, we introduce a novel framework that unifies the compression of long video tokens from two dimensions, as illustrated in Fig. 2:

1. *Token-level*: the **Learnable&Progressive Compression (LP-Comp)** represents each frame using as few as one token, which is learned via training (Sec. 3.2).
2. *Frame-level*: **Question-conditioned Compression (QC-Comp)** selects the frames that are most relevant to the questions based on the attention scores inside the LLM layers, which is employed as an inference-time enhancement (Sec. 3.3).

### 3.1 Preliminaries and Formulation

**VLMs for Video Understanding.** Current VLMs primarily follow the architecture of LLaVA [40] when handling visual inputs: (1) an encoder encodes the raw pixels into a set of visual tokens; (2) a projector, commonly several multi-layer perceptions (MLPs), projects the visual tokens into the dimensions the LLMs; (3) an LLM accepts both the text tokens and projected visual tokens and decodes the output in an auto-regressive manner. This paper concentrates on the compression between the LLM layers and keeps the visual encoder and projector unchanged.

More formally, we consider accepting videos as input, the standard practice is to sample $T$ frames for the VLM to process, denoted as $\{F_t\}_{t=1}^T$. Suppose the vision encoder converts each frame into $N^{(1)}$ visual tokens, we then let the LLM's input for the $i$-th frame be $V_i^{(1)} \in \mathbb{R}^{N^{(1)} \times d}$, where $d$ is the LLM's dimension. So the input video tokens for the LLM become a sequence of $T \times N^{(1)}$ tokens, namely, $V^{(1)} \leftarrow \left[V_1^{(1)}, ..., V_T^{(1)}\right]$. Denoting the input query as $Q^{(1)}$ with $N_q$ tokens in standard question-answering scenarios, we follow the practice of our baseline model VideoChat-Flash [35] of appending the query $Q^{(1)}$ after the visual tokens $V^{(1)}$, yielding the input sequence to LLM as $X^{(1)} \leftarrow [V^{(1)}, Q^{(1)}]$, a sequence with length $T \times N^{(1)} + N_q$. Then, the LLM begins the auto-regressive generation of new text tokens based on the input sequence $X^{(1)}$.

**Objective: Visual Token Compression.** The computation process of the LLM is commonly dominated by the large volume of visual tokens $T \times N^{(l)}$ in long video understanding. To reduce the cost of LLM layers and permit more input frames, we explore two directions to reduce video token counts: either from the aspect of $N^{(1)}$, the number of tokens per frame, or $T$, the number of frames. **(1)** *Token-level Compression*. Denoting $N^{(l)}$ as the number of tokens per frame at $l$-th transformer layer, our formal objective is to decrease $N^{(l)}$ so that the transformer layers have a gradually more concise set of visual tokens to process, shrinking from $T \times N^{(1)}$ to $T \times N^{(l)}$. We demonstrate such token-level compression in Sec. 3.2. **(2)** *Frame-level Compression*. Another way of decreasing the computation is to convert the frame number $T$ into a smaller $T'$. In the context of long video understanding, this means selecting a subset of relevant frames to process and ignoring the rest. Our

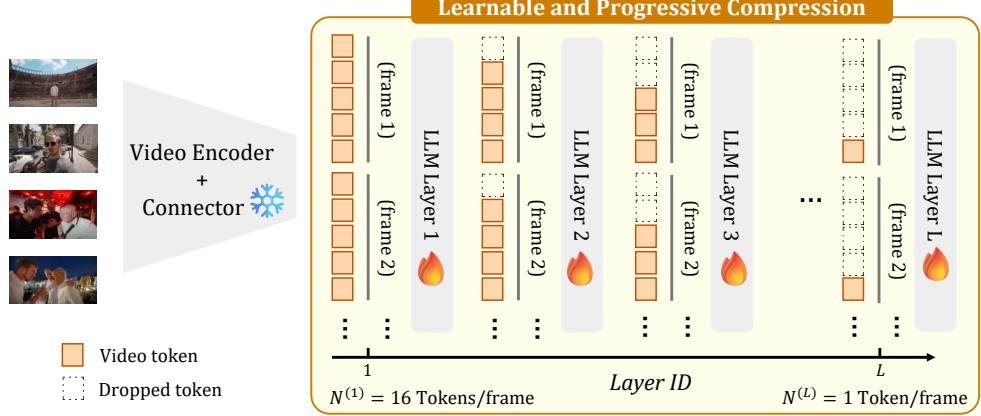

Figure 3: **Learnable and Progressive Compression (LP-Comp).** With *supervised compression tuning*, the LLM layers *learn* to condense the visual tokens *progressively* into a concise set of tokens until reaching the extreme of one token per frame.

Sec. 3.3 presents frame-level compression strategies. Finally, our framework unifies both types of compression to enhance long video understanding.

## 3.2 Learnable and Progressive Token Compression (LP-Comp)

**Objective.** As discussed in Sec. 3.1, the objective of *token-level compression* is to decrease the number of video tokens $N^{(l)}$ than the initial token number $N^{(1)}$. Suppose the LLM contains $L$ layers, we aim at enabling the LLM to compress the tokens so that *each frame is compressed to a single token* at the last layer, namely, $N^{(L)} = 1$. In the case of our baseline VideoChat-Flash [35], where $N^{(1)} = 16$, our target marks the compression ratio of 16.

Although previous studies have demonstrated the redundancy in visual modalities [5], especially for videos [37], such a level of compression is challenging, as modern VLMs like VideoChat-Flash have already utilized the projector for visual token compression in the form of downsampling. As a result, we discover that the heuristic compression approaches, *e.g.*, selecting tokens in a training-free manner, are prone to failure for such a large compression ratio (as in Sec. 4.3, Table 2), whose performance significantly falls behind the baseline without token compression. Therefore, our key insight is to let the LLM layers actively *learn* to condense the tokens into a concise set instead of purely relying on inference-time heuristics for extreme visual token compression.

**Learnable Token Compression.** As shown in Fig. 3, the LLM layers can be trained with the token compression by decreasing the number of tokens during the training stage. A critical design of such token compression is always to preserve the *suffix* tokens of a frame. Concretely, we assume a specific layer $k$ compresses visual tokens from $N^{(k)}$ to $N^{(k+1)}$, where $N^{(k+1)} < N^{(k)}$, then the first $N^{(k)} - N^{(k+1)}$ remaining tokens for a frame are determined to be removed (the dashed boxes in Fig. 3). Such a *suffix-preservation* behavior is compatible with the causal attention of decoder-only LLM layers, as the later tokens in a sequence can absorb the earlier token features instead of the reverse direction. By end-to-end supervising the LLM layers for compression, we observe a significant improvement over heuristics (Sec. 4.3, Table 2).

**Progressive Token Compression.** The effectiveness of learnable compression is determined by the capability of LLM layers. For instance, we could directly conduct extreme one-token-per-frame compression with the first LLM layer for maximum efficiency, *i.e.*, $N^{(2)} = 1$, but this would risk discarding critical visual information. To balance both token efficiency and preserving information, we propose to compress progressively across all the LLM layers. As illustrated in Fig. 3, each LLM layer only decreases a small amount of visual tokens according to a smooth schedule. We empirically find the cosine curve effective for token compression, where the visual token number evolves as,

$$N^{(\ell)} = \left\lceil \frac{N^{(1)} - 1}{2} \cos\left(\frac{\ell}{L}\pi\right) + \frac{N^{(1)} + 1}{2} \right\rceil, \quad \ell = 1, \ldots, L. \tag{1}$$

Our experiments (Sec. 4.3, Table 2) show that smoother compression indeed performs better.

**Data-efficient Supervised Compression Tuning.** We state such continued training of VLMs for the compression capability as *supervised compression tuning* (SCT). It can be learned in a data-efficient way: only using 2.5% of the data samples from VideoChat-Flash's SFT dataset. Besides, our learnable compression also improves the training efficiency by reducing the average number of tokens, accelerating the training speed, and enabling training on more frames.

**Discussion.** Despite its simplicity, we emphasize three critical insights leading to our effective token compression: (1) it is viable to conduct visual token compression between the LLM layers without only relying on the visual encoder and projector; (2) *learnable* LLM layers can grasp the condensing of visual information without only relying on training-free heuristics; (3) *progressive* schedule utilizes all the LLM layers for compression without only relying on a few selected layers.

### 3.3 Question-Conditioned Frame Compression (QC-Comp)

**Objective.** Recalling our objective in Sec. 3.1, this section addresses the compression at the frame level (from $T$ to $T'$). In its simplest form, our frame-level compression is achieved via frame selection, where only the frames relevant to the user's questions are used for the response. According to the prior works in VLMs, reducing such redundancy is beneficial to the performance and also enable us to digest more video frames. To complete such selection, we aim to assign a *relevance score* for each frame reflecting its usefulness to the user's question and then remove the ones with the lowest relevance; thus, this step is named *Question-Conditioned Compression*.

**QC-Comp Overview.** The QC-Comp is employed at the inference time for our model to select the most relevant frames. Similar to the previous works [6, 16, 50, 63], we also utilize the internal representation of the VLM for this objective. Concretely, we calculate the attention scores between the question tokens and the video tokens, which serve as the metric for the contributions of a frame: a larger attention score indicates better relevance, an intuition adopted by representative LLM [66] and video analysis [79]. However, previous approaches utilizing attention score for frame selection exhibit an under-explored issue: the bias of LLM attention for the tokens at the beginning and the end of a sequence, namely "*lost in the middle*" [42], which could overlook the critical information at the middle of videos. To address the above problem, we propose splitting a long video sequence into independent short clips.

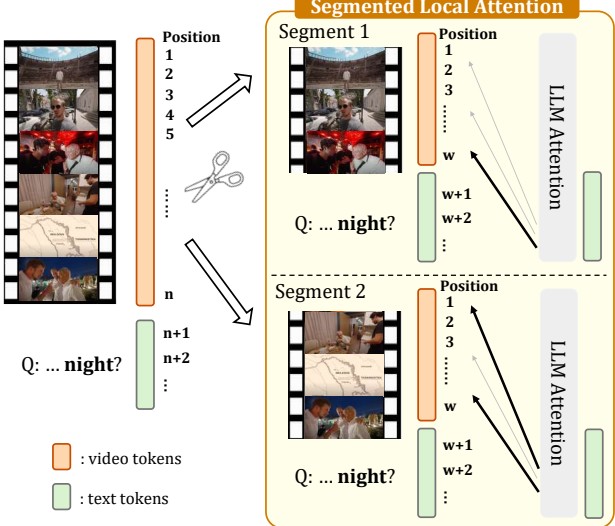

Figure 4: **Question-Conditioned Compression.** To reduce the visual redundancy and improve long video understanding performance, we split a video into individual segments and assign question-conditioned relevance scores to video frames. The frames with lower relevance scores are discarded so that the LLM only concentrates on the informative frames.

**Segmented Local Attention for Relevance Scoring.** The only change of our solution to conventional LLM layers is to split a video token sequence $V^{(l)}$ with $T$ frames into $N_S = \lceil (T-w)/\text{stride} \rceil$ segments, where $w$ is the window size of the segments. Denoting the $N_S$ segments as $\left[ \tilde{V}_1^{(l)}, ..., \tilde{V}_{N_S}^{(l)} \right]$, we compute the question-conditioned attention within each segment locally by connecting the segment-level video tokens with question tokens, *i.e.*, $[\tilde{V}_i^{(l)}, Q^{(l)}]$. The calculation of these segments is independent and can be conducted in parallel, as shown in Fig. 4. Finally, we record the average attention scores across different attention heads. With these attention scores, we calculate the relevant score for each frame as the average of attention score from any question token to any video token that represents this frame in any segment. Top-k relevant frames among the $T$ frames are selected.

Table 1: **Long video understanding comparison.** Our method XComp built upon VideoChat-Flash-2B achieves state-of-the-art results among 2B-scale models, demonstrating the effectiveness of extreme compression in long video understanding. Note that the performance of proprietary models and 3~9B-scale VLMs are provided for reference.

| Average Duration | Size | LongVideoBench 473s | MLVU 651s | VideoMME (Long) 2386s | LVBench 4101s |
|---|---|---|---|---|---|
| *Proprietary Models* | | | | | |
| GPT-4V [48] | - | 59.1 | 49.2 | 53.5 | – |
| GPT-4o [49] | - | **66.7** | **64.6** | 65.3 | 30.8 |
| Gemini-1.5-Pro [58] | - | 64.0 | – | **67.4** | **33.1** |
| *3~9B-Scale VLMs* | | | | | |
| Qwen2.5VL-3B [4] | 3B | 43.3 | 68.2 | – | – |
| mPLUG-Owl3 [73] | 7B | 52.1 | – | 50.1 | 43.5 |
| VideoChat-Flash-7B [35] | 7B | 64.7 | 74.7 | **55.4** | **48.2** |
| Eagle2.5-8B [8] | 8B | **66.4** | **77.6** | – | – |
| Kangaroo [41] | 8B | 54.8 | 61.0 | 46.7 | 39.4 |
| TimeMarker [10] | 8B | 56.3 | – | 46.4 | 41.3 |
| InternVL3-9B [81] | 9B | 62.5 | 70.8 | – | – |
| *2B-Scale VLMs* | | | | | |
| InternVL3-2B [81] | 2B | 55.4 | 64.2 | – | – |
| VideoChat-Flash-2B [35] | 2B | 58.3 | 65.7 | 44.9 | 42.9 |
| XComp | 2B | **59.7** | **66.7** | **45.6** | **46.2** |

**Discussion.** Despite the complexity of duplicating the question token, it is viable to mitigate the bias in long-context LLM attention, resulting in an improved performance as shown in Table 4.

## 4   Experiments

### 4.1   Implementation Details

We fine-tune XComp from VideoChat-Flash-2B [35] for supervised compression tuning, integrating our proposed LP-Comp mechanism. The core components follow the original setup from VideoChat-Flash: we use UMT-L [43] as the visual encoder, token merging with an MLP-based connector, and Qwen2-1.5B [71] as the large language model (LLM). For long videos, we segment them into short clips of 8 frames each. Each clip is compressed into 128 visual tokens, leading to an average of 16 tokens per frame.

Our LP-Comp mechanism introduces a progressive token reduction strategy across LLM layers. Initially, each frame is represented by 16 tokens. After each LLM layer $l$, we compute the target token count $N^{(l)}$ as defined in Equation 1. If the token count from the previous layer $N^{(l)}$ differs from $N^{(l-1)}$, we uniformly drop $N^{(l-1)} - N^{(l)}$ tokens for each frame across the temporal dimension. This process continues across all layers, eventually reducing to a single token per frame at the final LLM layer, *i.e.*, $N^{(L)} = 1$. With LP-Comp reducing the token sequence length during training, our method enables faster training of VLMs on video sequences with more frames. Concretely, we use the mixture of 128–1024 frames per video with up to 8 frames per second in fine-tuning. The fine-tuning data follows the mixture of short-video and long-video instruction tuning formats used in VideoChat-Flash [35], but our supervised compression tuning only requires 2.5% of VideoChat-Flash's dataset. The training process costs around 24 hours on 8×NVIDIA H100 GPUs.

During the inference stage, we utilize QC-Comp to conduct frame selection. When splitting the long video into short segments, each segment contains 64 frames, and we select 512 frames for videos with a duration of less than an hour, and 2048 frames for videos longer than an hour. More details on implementation are in the supplementary materials.

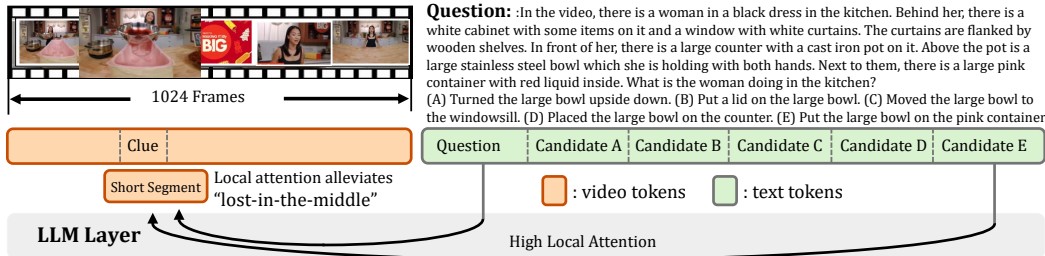

Figure 5: **LongVideoBench case analysis.** XComp leverages QC-Comp to divide a long video into short segments, mitigating attention bias. Local attention highlights key frames relevant to the question and correct answer, enabling effective filtering of irrelevant content.

## 4.2 Main Results on Long Video Understanding

**Benchmarks.** We assess our model on four widely used long video understanding benchmarks: LongVideoBench [64], MLVU [80], LVBench [59], and VideoMME [18] (w/o subtitles). Unlike short-video datasets, they feature extended video durations ranging from several minutes to over an hour, posing greater temporal reasoning and memory challenges. These benchmarks adopt a question-answering format, primarily using accuracy as the evaluation metric.

**Comparison.** We choose the baselines with similar model size of VideoChat-Flash-2B [35] and InternVL3-2B [81], as shown in Table 1. Importantly, our XComp is trained from VideoChat-Flash using only 2.5% of the data, the improvement over VideoChat-Flash suggests that our *one token per highly selective frame* preserves the critical information after compression.

**Leading Performance.** As shown in Table 1, our XComp consistently improves the baseline VideoChat-Flash and outperforms the other VLMs at similar scales, even exhibiting comparable performance to some models at 7B scale. As also demonstrated in Fig. 1, our XComp demonstrates the unique trend of improving performance with more input frames, demonstrating the effectiveness of our extreme compression.

**Qualitative Analysis.** Figure 5 presents an example from LongVideoBench illustrating how XComp leverages QC-Comp to effectively identify frames relevant to the question. In this long video question answering scenario, the question provides a detailed description of the background and inquires about the character's actions. Five answer candidates are also included. To answer the question, QC-Comp divides the video into shorter segments of 64 frames each, mitigating attention biases such as the "lost-in-the-middle" issue. This segmentation allows the model to focus local attention on key frames, which receive high relevance scores with respect to both the question and the correct answer candidate. As a result, XComp is able to discard 512 irrelevant frames and accurately identify the correct answer.

## 4.3 Ablation Study

**Learnable Compression.** The key insight of our XComp is to train the LLM layers to condense the visual tokens. In Table 2, we compare using learnable with training-free token compression. Enabling the LLM layers to learn the compression behavior consistently outperforms the training-free model, supporting the necessity of our learnable compression. Another critical comparison is that when using the heuristic token compression from VideoChat-Flash, which utilizes attention scores to select the tokens, it is worse than the original baseline under our extreme token

Table 2: **Ablation study using LVBench:** progressive v.s. step-wise non-heuristic token compression v.s. heuristic compression from VideoChat-Flash, and learnable v.s. training-free token compression. All the evaluations are conducted with 1024 frames. Both progressive compression and staged compression would gradually compress tokens to 1 token per frame with the same amount of average tokens per frame.

| Method | Training-free | Learnable |
|---|---|---|
| Baseline (w/o compression) | 41.8 | - |
| + Heuristic | 41.1 | - |
| + Step-wise | 38.4 | 42.3 |
| + Progressive | 39.7 | **44.3** |

compression ratio. This is precisely the observation that motivates us to propose the learnable token compression (Sec. 3.2).

**Progressive Compression.** To investigate the importance of progressive token compression, *i.e.*, utilizing all the LLM layers to gradually decreasing the token number, we conduct ablation experiments comparing it with the conventional *step-wise* approach, which performs abrupt token reduction at pre-defined layers as in VideoChat-Flash [35]. As shown in Table 2, progressive token compression consistently outperforms the step-wise compression, indicating that smooth compression schedule is essential for preserving the critical visual information.

Table 3: **Comparison of token compression strategies on LVBench and MLVU.** Both Uniform Drop and Suffix-Preservation follow the same Cosine compression schedule and efficient continual training.

| Strategy | LVBench | MLVU |
|---|---|---|
| Uniform | 43.6 | 64.1 |
| Suffix | 44.3 | 65.2 |

**Compressing to Suffix Tokens.** A critical design choice in our LP-Comp is that the LLM layers should gradually preserve the information to the last token of each frame, namely, the suffix tokens. Such a design is compatible with the causal attention in LLM layers, and we analyze its necessity by comparing it with another intuitive strategy: keeping the tokens uniformly by their positions (implementation details in the supplementary materials). As shown in Table 3, preserving the suffix tokens outperforms its uniform counterpart strategy, supporting the importance of designing the compression strategy aligning with the attention of LLM layers.

**Segmented Local Attention.** To understand the impact of segment-wise attention in QC-Comp, which is proposed to mitigate the position bias of attention in long contexts, we compare it with the conventional way of computing the attention between the question and all frames globally. As shown in Table 4, segment-wise attention consistently leads to better performance across all three long video QA benchmarks LongVideoBench, MLVU, and VideoMME (long). This validates our design choice of computing frame scores locally within segments, effectively reducing positional bias and improving the relevance of retained frames in the early filtering stage.

Table 4: **Comparison of attention strategies in QC-drop.** Segment-wise attention consistently outperforms global attention across benchmarks.

| Attention | LongVideoBench | MLVU | VideoMME (long) |
|---|---|---|---|
| Global | 59.5 | 66.3 | 44.8 |
| Segment Local (Ours) | 59.7 | 66.7 | 45.7 |

**Effect of LP-Comp and QC-Comp.** We perform an ablation study to analyze the impact of the two components in XComp: LP-Comp and QC-Comp. As shown in Table 5, the baseline without any compression achieves a score of 58.3 on LongVideoBench. Adding LP-Comp alone (XComp without QC-Comp) yields a slight improvement to 58.8. When both LP-Comp and QC-Comp are used (full XComp), the performance reaches 59.7, suggesting that selectively removing less relevant content may help retain useful information. These results indicate that both components contribute to the final performance.

Table 5: **Ablation results showing the individual and combined impact of LP-Comp and QC-Comp on LongVideoBench.** Both components contribute to the overall performance gains of XComp.

| | LongVideoBench |
|---|---|
| Baseline (w/o compression) | 58.3 |
| XComp w/o QC-Comp | 58.8 |
| XComp | 59.7 |

Table 6: **The complete table of all ablation studies on LongVideoBench and LVBench.**

| Model Name | LP-Comp | Variants | QC-Comp | Variants | LongVideoBench | LVBench |
|---|---|---|---|---|---|---|
| VideoChat-Flash | No | - | No | - | 58.3 | 42.9 |
| XComp | Yes | Uniform | No | - | 58.2 | 43.6 |
| XComp | Yes | Suffix | No | - | 58.8 | 44.3 |
| XComp | Yes | Suffix | Yes | Global | 59.5 | 45.6 |
| XComp | Yes | Suffix | Yes | Segment Local | **59.7** | **46.2** |

**Compare All Variants Together.** Table 6 shows the complete results for all ablation studies of XComp design, on two benchmarks, LongVideoBench and LVBench. The results are consistent with the studies above.

## 4.4 XComp for LLaVA-Next-Video

We apply XComp to LLaVA-Next-Video [78] to demonstrate the generalizability of our design. Similar to the experiments reported in the paper, XComp effectively enhances model efficiency, allowing a larger maximum number of frames to be processed under limited GPU memory. Following the paper's setup, we use only 2.5% of the training data of LLaVA-Next-Video (i.e., LLaVA-Video-178k).

For a fair comparison, we also evaluate LLaVA-Next-Video with the same amount of fine-tuning data while keeping the model architecture unchanged. As shown in Table 7, XComp improves performance across both benchmarks.

Table 7: **Generalize to LLaVA-Next-Video, a different model from VideoChat-Flash.** XComp consistently improves the efficiency and accuracy of LLaVA-Next-Video on the two long video benchmarks, outperforming both the baseline and fine-tuning.

| Model Version | VideoMME (Long) | LVBench |
|---|---|---|
| LLaVA-Next-Video | 62.7 | 40.6 |
| LLaVA-Next-Video + FT | 61.4 | 41.4 |
| LLaVA-Next-Video + XComp | **63.2** | **43.1** |

## 5 Conclusion

In this work, we propose extreme token compression, named **XComp**, to address the long video understanding problem by alleviating the large number of tokens. At the token level, our XComp framework targets on compressing each frame to *one token*. Observing the failures of heuristic compression adopted by previous methods, we propose the key insight of letting the LLM layers *learn* the *progressive* compression of video tokens. This compression strategy, called LP-Comp, effectively improves the token efficiency and enables denser frame sampling. Our compression at the frame level further enhances the information contained in the tokens by selecting the frames relevant to the user queries. Based on the internal attention scores of LLM layers, we further propose to split the whole video into shorter segments to mitigate the position bias issues. The resulting *question-conditioned* compression, QC-Comp, curates a selective set of frames for the LLM to process. Collectively, our proposed strategies enable XComp to explore the extreme of video token compression via data-efficient fine-tuning and significantly improve the base VideoChat-Flash on multiple long video understanding benchmarks.

**Limitations and Future Work.** Under a limited budget, we primarily explore the extreme token compression with VideoChat-Flash-2B and LLaVA-Next-Video via data-efficient fine-tuning. Therefore, potential future work includes adapting our XComp framework to larger VLMs and potentially integrating the learnable compression into the pre-training stage of the VLM. In addition to the scale and diversity of experiments, another future work lies in training with the question-conditioned frame selection, which could be jointly optimized with the LLM layers to achieve even larger compression ratios than presented in the paper.

**Broader Impacts.** This work presents advancements in vision-language modeling, with a focus on improving long video understanding. While our contributions are primarily foundational, we acknowledge potential societal risks associated with vision-language models, including misuse of disinformation, privacy concerns, and the amplification of biases present in training data. We encourage future work to include fairness assessments and responsible release strategies. Where applicable, safeguards should be considered to mitigate unintended harms.

## Acknowledgments

This work was supported in part by Amazon, NSF under Grants 2106825 and 2519216, the ONR Grant N00014-26-1-2099, and the DARPA Young Faculty Award. This work used computational resources, including Amazon Web Services (AWS), and the NCSA Delta and DeltaAI supercomputers through allocation CIS230012 from the Advanced Cyberinfrastructure Coordination Ecosystem: Services & Support (ACCESS) program.

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

# A  Implementation Details

## A.1  XComp Architecture

**XComp** is based on VideoChat-Flash [35] and uniquely integrates our learnable progressive compression (**LP-Comp**)(Sec. 3.2) and question-conditioned compression (**QC-Comp**)(Sec. 3.3). XComp comes from fine-tuning the VideoChat-Flash-2B model [35] with a small amount of 2.5% data, the fine-tuning details are shown in Appendix A.2.

Algorithm A shows the neural network architecture of XComp, which is largely consistent with VideoChat-Flash. It begins with the UMT-L visual encoder [43], which encodes short video clips consisting of 8 frames into visual tokens. Token merging [5] is subsequently applied to reduce the token count to 128 tokens per clip and 16 tokens per frame. A two-layer MLP connector then maps these visual tokens from the visual encoder space into the representation space of the large language model. Finally, Qwen2-1.5B [4] serves as the large language model (LLM) in our framework.

During inference with QC-Comp, multiple forward passes through the model are performed. First, XComp conducts a forward pass to obtain scores for the frames. Note that for videos longer than 512 frames, the video is divided into shorter segments, each processed separately to compute frame scores. These individual scores are then aggregated. Based on the aggregated scores, XComp selects the top-ranked frames and passes them to the model to generate responses.

**Details of LP-Comp**  Algorithm B shows the details of LP-Comp (Sec. 3.2). It compresses video tokens layer-by-layer in a suffix-preserving manner. Given input tokens $V^{(\ell)} \in \mathbb{R}^{K \times N^{(\ell)} \times d}$ at layer $\ell$, the algorithm computes the target token count $N_{\text{next}}$ for the next layer using a cosine-based schedule. If no reduction is needed, tokens are returned unchanged. Otherwise, for each video clip and each frame within it, only the last $N_{\text{next}}$ tokens are kept from the current $N_{\text{prev}}$, preserving the temporal suffix. The retained tokens across all frames and clips are concatenated to form the output $V^{(\ell+1)}$. This design ensures progressive compression while retaining semantically rich information.

**Details of QC-Comp**  Algorithm C shows the details of Segmented Local Attention, which is the main part of QC-Comp (Sec. 3.3) and computes scores from attention maps in LLM layers. For a given video $V$ and text query $Q$, the algorithm slides a local window (64 frames with a stride of 32) over the video sequence. Within each segment, it computes the full attention map from the LLM's $\ell$-th layer. For each frame in the segment, the attention weights over the query tokens are averaged and accumulated into corresponding 8-frame clip buckets. Although Segmented Local Attention inherently produces clip-level scores, QC-Comp assigns the same score to all frames within a clip, thereby converting clip-level scores to frame-level scores. For long videos exceeding 512 frames, the

---

**Algorithm A:** MODEL_FORWARD: forward pass with LP-Comp and the score for QC-Comp

**Input:** $\mathcal{M}$ (model), $V$ (video), $Q$ (text), $returnScore \in \{0, 1\}$
**Output:** $logits$; $score$ (per-clip) if $returnScore = 1$
**/* Step 1: video encoding */**
$\{V_1^{\text{clip}}, \ldots, V_K^{\text{clip}}\} \leftarrow \texttt{Partition}(V, 8)$         `// separate each 8 frames → 1 clip`
**foreach** $V_k^{\text{clip}}$ **do**
    $T_k \leftarrow \mathcal{M}.\text{umt\_visual\_enc}(V_k^{\text{clip}})$
    $T_k \leftarrow \texttt{TokenMerge}(T_k, 128)$         `// to 128 tokens/clip, 16 tokens/frame`
    $T_k \leftarrow \mathcal{M}.\text{mlp}(T_k)$         `// project to LLM dim`
$V^{(0)} \leftarrow \text{concat}(T_1, \ldots, T_K)$
$Q^{(0)} \leftarrow \mathcal{M}.\text{text\_embed}(Q)$
**/* Step 2: reasoning in LLM layers */**
**for** $\ell = 0$ **to** $\mathcal{M}.L - 1$ **do**
    $[V^{(\ell+1)}, Q^{(\ell+1)}] \leftarrow \mathcal{M}.\text{llm\_layer}_\ell(V^{(\ell)}, Q^{(\ell)})$
    $V^{(\ell+1)} \leftarrow \texttt{LP\_Comp}(\ell, V^{(\ell+1)})$         `// token-level compression`
    $score[\ell] \leftarrow \texttt{SegmentedLocalAttention}(\ell, V^{(\ell)}, Q^{(\ell)})$     `// score for QC_Comp`
$logits \leftarrow \texttt{ComputeLogits}(Q^{(\mathcal{M}.L)})$
**if** $returnScore$ **then return** $(logits, score)$
**else return** $logits$

---

method processes multiple overlapping 512-frame chunks. The final frame-level scores are obtained by aggregating the results from these overlapping chunks. To ensure diverse frame-level scores, each frame would be included in $n_{\text{repeat}}$ overlapping and shifted chunks that encourage score variation across neighboring frames. With these frame-level scores, $n\_selected\_frames$ among total frames are selected. Table B shows the hyperparameters.

## A.2 Supervised Fine-tuning Details

**Hyperparameters** Table A shows the hyperparameters used in fine-tuning. We follow the same training configuration as VideoChat-Flash [35], including the learning rate, weight decay, warmup ratio, and learning rate scheduler. The only difference lies in the frame sampling parameters: `frames_upbound`, `frames_lowbound`, and the default frames per second (FPS). This change is due to our use of LP-Comp, which enables more efficient frame representation. As a result, we double the values of `frames_lowbound` and `frames_upbound` compared to VideoChat-Flash.

**Datasets** We used the 2.5% supervised fine-tuning data collected or released by VideoChat-Flash [35]. During data curation, we disregarded a few datasets that were exceptionally large on disk or whose licenses made automatic download impractical. After this filtering, the final training set contains 71,927 instances are drawn from publicly available datasets, listed in Table C.

---

**Algorithm B:** LP_COMP: suffix-preserving layer-wise video-token compression

**Input:** $\ell$ (layer index), $V^{(\ell)}$ (video tokens, shape $K \times N^{(\ell)} \times d$)
**Output:** $V^{(\ell+1)}$ (compressed video tokens)

$N_{\text{prev}} \leftarrow \left\lceil \frac{N^{(1)}-1}{2} \cos\left(\frac{\ell}{L}\pi\right) + \frac{N^{(1)}+1}{2} \right\rceil$

$N_{\text{next}} \leftarrow \left\lceil \frac{N^{(1)}-1}{2} \cos\left(\frac{\ell+1}{L}\pi\right) + \frac{N^{(1)}+1}{2} \right\rceil$  // Eq.(1)

**if** $N_{\text{prev}} = N_{\text{next}}$ **then return** $V^{(\ell)}$  // nothing to compress

**foreach** $T \in V^{(\ell)}$ **do**  // iterate over $K$ clips
  **foreach** $f \leftarrow 1$ **to** $F$ **do**  // iterate over $F$ frames in the clip
    idx_keep $\leftarrow \left\lceil (f-1)N_{\text{prev}} + N_{\text{prev}} - N_{\text{next}}, \ldots, fN_{\text{prev}} - 1 \right\rceil$
    $T'_f \leftarrow T[\text{idx\_keep}]$  // suffix-preservation
  $T' \leftarrow \text{concat}(T'_1, \ldots, T'_F)$
$V^{(\ell+1)} \leftarrow \text{concat}(T'\_\text{clip1}, \ldots, T'\_\text{clipK})$
**return** $V^{(\ell+1)}$

---

**Algorithm C:** SEGMENTEDLOCALATTENTION: compute clip-level saliency scores

**Input:** $\ell$ (layer index), $V$ (video tokens), $Q$ (text tokens)
**Output:** $score$ (list of length $K$, one mean score per clip)

$L_{\text{seg}} \leftarrow 64$  // frames per segment (8 clips)
stride $\leftarrow 32$  // stride in frames (4 clips)
$bucket \leftarrow [\ ]\_k = 1^K$
**for** $start = 0$ **to** $F - L_{\text{seg}}$ **step** $stride$ **do**  // slide local window over the video
  $V_{\text{seg}} \leftarrow V[start : start + L_{\text{seg}}]$
  $A \leftarrow \mathcal{M}.\text{llm\_layer}_\ell.\text{attn}([V_{\text{seg}}, Q])$  // full attention map
  **for** $i = 0$ **to** $L_{seg} - 1$ **do**
    $c \leftarrow \left\lfloor \frac{start+i}{8} \right\rfloor$  // clip index
    $w \leftarrow \text{mean}(A[i, Q\_start : Q\_end])$
    $bucket[c].\text{append}(w)$

**for** $k = 1$ **to** $K$ **do**
  $score[k] \leftarrow \text{mean}(bucket[k])$  // final clip score
**return** $score$

Table A: **The hyperparameters used in fine-tuning.**

| Hyperparameter | Value / Description |
|---|---|
| tunable_parts | Large Langauge Model |
| learning_rate | $1 \times 10^{-5}$ |
| weight_decay | 0.0 |
| warmup_ratio | 0.03 |
| lr_scheduler_type | Cosine |
| dataloader_num_workers | 1 |
| frames_upbound | 1024 |
| frames_lowbound | 128 |
| local_num_frames | 8 |
| sample_type | Dynamic FPS (8 fps by default) |

Table B: **The hyperparameters used in evaluation.**

| Hyperparameter | Value / Description |
|---|---|
| temperature | 0.0 |
| do_sample | False |
| num_beams | 1 |
| n_repeat | 2 (each frame in 2 chunks) |
| n_selected_frames | 256 (Long), 512 (MME), 1,024 (MLVU), 2,048 (LVB) |

Table C: **Datasets used for supervised fine-tuning (71,927 instances).**

| Image datasets | Video datasets |
|---|---|
| LLaVA-OneVision [32], LLaVA-NeXT [39], M4-Instruct [33] | Kinetics-400 [27], Something-Something [20], TGIF-QA [25], TVQA [30], CLEVRER [74], NExT-QA [67], FAVD [54], MovieChat-1K [57], TextVR [65], ShareGPT-Video [9], ShareGPT-4o [14], Oops [15], OVIS [51], UVO [60], GUI-World [7], Vript [72], HT-Step [1], Ego4D [21], LLaVA-Video-178K [78], VideoChat-Flash [35] |

# B Additional Experiments

## B.1 Ablation Study on Fine-tuning

Table D presents an ablation study to isolate the effect of our proposed design from that of fine-tuning. Specifically, we compare XComp with a baseline that applies the same fine-tuning procedure (on 2.5% of the data as mentioned in Appendix A.2) to the original VideoChat-Flash-2B model, without introducing our LP-Comp and QC-Comp. VideoChat-Flash-2B+FT achieves comparable performance to the original model, suggesting a limited benefit from fine-tuning. This indicates that the performance gains of XComp stem from our method's enhancements, rather than from fine-tuning alone.

Table D: **Ablation study of fine-tuning.**

| | Size | LongVideoBench | MLVU | VideoMME (Long) | LVBench |
|---|---|---|---|---|---|
| Average Duration | | 473s | 651s | 2386s | 4101s |
| VideoChat-Flash-2B [35] | 2B | 58.3 | 65.7 | 44.9 | 42.9 |
| VideoChat-Flash-2B+FT | 2B | 57.4 | 65.6 | 44.7 | 43.2 |
| XComp | 2B | **59.7** | **66.7** | **45.6** | **46.2** |

## B.2 Efficiency Analysis

Table E shows the efficiency comparison on a LVBench query (863 text tokens), measured over 10 runs on a single NVIDIA H200. Across 1k–4k frames, XComp reduces LLM TFLOPs by ∼53–58% and latency by ∼45–58%.

Table E: **Efficiency Comparison on LVBench Queries at Inference Stage.**

| Frames | Model | TFLOPs | | Latency | |
|--------|-------|--------|------|---------|------|
| 1,024 | VideoChat-Flash | 43 | | 0.22 s | |
| 1,024 | Ours | 20 | (**53%↓**) | 0.12 s | (**45%↓**) |
| 2,048 | VideoChat-Flash | 132 | | 0.54 s | |
| 2,048 | Ours | 58 | (**56%↓**) | 0.25 s | (**54%↓**) |
| 4,096 | VideoChat-Flash | 448 | | 1.56 s | |
| 4,096 | Ours | 187 | (**58%↓**) | 0.66 s | (**58%↓**) |

## B.3 More Evaluations

While XComp primarily targets multiple-choice tasks on long videos, it is also capable of handling other fundamental video understanding tasks. We claim that the extreme compression is not harmful to the fundamental capability for short videos. In particular, we include evaluations on (1) reasoning benchmarks such as CLEVRER, which test causal and counterfactual reasoning, and (2) dense-output tasks such as video captioning (VDC benchmark), which assess the ability to generate descriptive text for videos. As shown in Table F, XComp achieves results comparable to its backbone model (VideoChat-Flash-2B), demonstrating that the compression does not significantly harm performance on these tasks.

We additionally evaluate on MME-VideoOCR, a benchmark designed to measure fine-grained text perception from videos. Table G reports results on both the full benchmark and the subset of long videos (>30s). XComp shows a moderate performance drop compared to the original VideoChat-Flash-2B, consistent with our main paper observations that this is largely due to data differences rather than architecture changes. When compared to fine-tuned with the same backbone, the performance gap is minimal, confirming that the degradation primarily comes from training data rather than compression.

Table F: **CLEVRER and VDC results.**

| | CLEVRER | | | | | | VDC | |
|---|---|---|---|---|---|---|---|---|
| | Expl (Opt) | Expl (Q) | Pred (Opt) | Pred (Q) | Cntrf (Opt) | Cntrf (Q) | BLEU@1 | BLEU@4 |
| VideoChat-Flash-2B | 0.9463 | 0.8497 | 0.7789 | 0.5738 | 0.7775 | 0.4007 | 7.1 | 1.6 |
| XComp | 0.9398 | 0.8406 | 0.7697 | 0.5600 | 0.7625 | 0.3652 | 7.0 | 1.6 |

Table G: **MME-VideoOCR results.**

| Model | MME-VideoOCR (Overall) | MME-VideoOCR (>30s) |
|-------|------------------------|---------------------|
| VideoChat-Flash-2B | 37.1 | 49.1 |
| VideoChat-Flash-2B + FT | 34.9 | 46.0 |
| XComp | 35.4 | 46.4 |

## B.4 Multi-hop NIAH

Figure A shows the results of the Multi-Hop Needle-in-a-Haystack QA task [35] which is designed to evaluate extreme long-context reasoning abilities. This benchmark embeds a reasoning path of images within long video sequences, where each image contains clues guiding the model to the next. Given a starting point, the model must trace the correct path, identify the target image (needle), and answer a related question. In this experiment, we use QC-Comp with `n_selected_frames = 1`

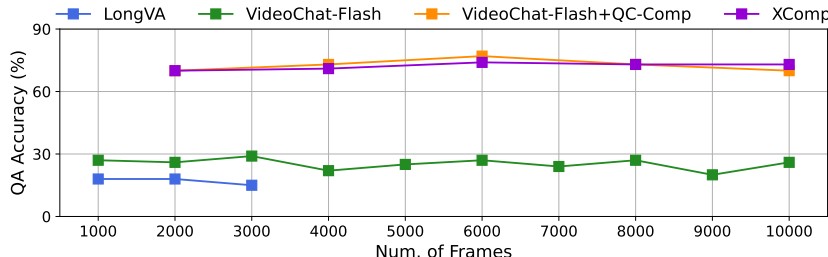

Figure A: **Multi-Hop Needle-in-a-Haystack QA Performance.**

and `n_repeat` $= 8$. It accurately selects the keyframe with 65% accuracy at a sequence length of 6,144, leading to QA average accuracies of 72.6 and 72.2 over 2,000 to 10,000 total frames for VideoChat-Flash+QC-Comp and XComp, respectively.

