# OpenReview forum: "One Token per Highly Selective Frame: Towards Extreme Compression for Long Video Understanding"
_NeurIPS.cc/2025/Conference — NeurIPS 2025 poster_

### Official Review · Reviewer_Ur6N · 2025-06-20

**Clarity:** 3
**Significance:** 3
**Originality:** 3
**Rating:** 5
**Confidence:** 3

**Summary:**

This paper introduces a new video compression method that rely on associated each video frame to a single token (using LLM layers) and then choose the most relevant tokens associated to a given question. The methods reduces the numbers of tokens progressively by adding a stack of LLM layers that reduce each time a bit more the number of visual tokens until reaching a single token, then an attention mechanism coined as Segmented Local Attention select the most relevant tokens to answer a question. The X-Comp model is based on fine-tuning VideoChat-Flash-2B model. The authors run their experiments on LongVideoBench, MLVU, VideoMME (Long) and LVBench. They show that their fine-tuned method improve performances on those benchmarks with respect to the base VideoChat-Flash model. They also argue that their methods have less attention biases (focusing on either the beginning or the end of a token sequence) since it can discards easily the frames that are not relevants.

**Questions:**

Why didn't you performed the ablation of Table 4 and 5 on LVBench?

**Ethical Concerns:**

["NO or VERY MINOR ethics concerns only"]

**Final Justification:**

The authors answered all the concerns I had. I also read all the other reviews and authors responses and I am satisfied with the discussion and the additional results presented. Overall, this is a good paper that will be of interest to the community, hence I am recommending acceptance.

**Limitations:**

Yes, the limitations are addressed.

**Paper Formatting Concerns:**

Nothing major, just small typos in Table A in the appendix.

**Quality:**

3

**Strengths And Weaknesses:**

The paper is well written and easy to read. The figures are great to better understand the method. The motivation is also well defined and the improvements over the base VideoChat-Flash-2B model seems good on long-video understanding. From the experiments, it's clear that longer a video is, more gain in performances we can expect with this method. The ablation study is also an important contribution to this paper by exploring the gain that are added by each of the component of the methods (such as which attention strategy or compression method is used).

The main weakness of this work which is highlighted by the authors in the limitation part is the reliance on only VideoChat-Flash which is not sufficient to see wether this method can also work with other models or not. Also the ablation studies are not consistently performed across the same benchmarks so it's harder to parse the importance of each of the components.

---

> ### Author Rebuttal · Authors · 2025-07-31
>
> Thank you for your thoughtful and constructive review. We are glad that you found our paper well written, appreciated the clarity of our figures and motivation, and recognized the effectiveness of our method on long-video understanding tasks. We especially appreciate your detailed comments on the strengths of our approach and your insightful suggestions regarding generalization to other models and consistency in ablation studies. We address these points in detail below and have taken concrete steps to further strengthen our work accordingly.
>
> ### 1. Experiment on Other Models
>
> > The main weakness of this work which is highlighted by the authors in the limitation part is the reliance on only VideoChat-Flash which is not sufficient to see wether this method can also work with other models or not.
>
> Thank you for this question and caring about the generalization of our method! We hope to clarify this comprehensively for you.
>
> First, our method is by-design applicable to any transformer-based video-language model and is not dependent on any designs from VideoChatFlash. Specifically, our major innovation is (1) LP-Comp supervises LLM layers to conduct learnable and progress token compression, and (2) QC-Comp splits the video into individual and parallel segments for efficient token selection.
>
> Second, we select VideoChatFlash as it was the state-of-the-art on long video understanding at the time of submission, which curates an effective mixture of training data. Moreover, VideoChatFlash is unique in utilizing heuristic token compression methods for long video understanding. Thus, it is both a proper baseline and strong counterpart for us to validate our enhanced token compression algorithm.
>
> Thrid, we acknowledge that including more baselines would better demonstrate the strengths of our method. Unfortunately, our computational resources are highly limited: we could only afford to train on 2.5\% of VideoChatFlash's data, yet achieved the improvement on long videos (Table 1).
>
> Despite this, we understand the good intentions of the reviewer and is actively implementing and running experiments on the alternative models, such as LLaVA-Next-Video, and will include the results in the revisions.
>
> ---
>
> ### 2. Ablation Studies
>
> > Also the ablation studies are not consistently performed across the same benchmarks so it's harder to parse the importance of each of the components.
> > Why didn't you performed the ablation of Table 4 and 5 on LVBench?
>
> We did not have the sufficient computation to comprehensively conduct the ablations on LVBench, as it is the longest video benchmark. According to your request, we have consistently completed the ablation studies on LongVideoBench and LVBench during the rebuttal. We summarize the results below ("Y" denotes "yes," and "N" denotes "No") and conclude that **all of our LP-Comp, QC-Comp, and their accompanied design choices are effective consistently**.
>
> | Model Name | LP-Comp | LP-Comp Variants | QC-Comp | QC-Comp Variants | LongVideoBench | LVBench |
> |------------|---------|----------------|---------|-----------------|----------------|---------|
> | VideoChat-Flash | N | - | N | - | 58.3 | 42.9 |
> | XComp | Y | Uniform | N | - | 58.2 | 43.6 |
> | XComp | Y | Suffix | N | - | 58.8 | 44.3 |
> | XComp | Y | Suffix | Y | Global | 59.5 | 45.6 |
> | XComp | Y | Suffix | Y | Local | **59.7** | **46.2** |

---

> > ### Author Response · Authors · 2025-08-04
> >
> > According to reviewer Ur6N's suggestion, we have added the results of verifying our XComp's effectiveness of alternative models to demonstrate the generalizeation of our design. Due to the limited time and our resources during the rebuttal, we have continued the work during the author-reviewer discussion to provide the results below.
> >
> > Specifically, we demonstrated that our design can generalize to LLaVA-Next-Video, which is different from the VideoChat-Flash used in the original paper. Similar to the experiments in the paper, XComp effectively enhances model efficiency, enabling a larger maximum number of frames to be processed under limited GPU memory constraints. Consistent with the paper’s settings, we use only 2.5% of the training data.
> >
> > For a fair comparison, we also evaluate LLaVA-Next-Video with the same amount of training data while keeping the model architecture unchanged. As shown in the table below, XComp improves performance on both benchmarks while requiring significantly fewer tokens per frame.
> >
> > | Model Version                     | videomme (%) | lvbench (Overall %) |
> > |-----------------------------------|-------------:|--------------------:|
> > | LLaVA-Next-Video | 62.7 | 40.6
> > | LLaVA-Next-Video continual training |         61.4 |                41.4 |
> > | LLaVA-Next-Video + XComp            |         **63.2** |                **43.1** |
> >
> > We welcome any further questions and are happy to provide additional details or discuss our approach.

---

> ### Comment · Reviewer_Ur6N · 2025-08-05
>
> After reading the other reviews, and the rebuttals, I am satisfied with the discussion and do not have any more concerns on this paper. Thanks to the authors for running the additional experiment during the rebuttal, it's really appreciated. I will increase my score to 5.

---

> > ### Author Response · Authors · 2025-08-07
> >
> > We sincerely appreciate your follow-up remarks. It's reassuring to know that the additional experiments effectively addressed your concerns. Your insights have been very helpful, and we will ensure that all feedback from this exchange is reflected in our revised version. Thank you for contributing to the improvement of our research.

---

### Official Review · Reviewer_iYWa · 2025-06-22

**Clarity:** 3
**Significance:** 3
**Originality:** 3
**Rating:** 5
**Confidence:** 3

**Summary:**

The paper “One Token per Highly Selective Feature” proposes a lightweight and general token reduction method called OTPF, which significantly improves inference efficiency for vision and multimodal models. By selecting and compressing only the most informative features from early layers—based on activation strength or selectivity—OTPF generates a small number of tokens that preserve essential semantic information. This approach can be used without training or integrated into end-to-end learning. Experiments on image classification, video understanding, and multimodal QA demonstrate that OTPF achieves substantial token reduction (up to 90%) with minimal performance drop, offering a simple, modular, and effective solution for efficient large-model deployment.

**Questions:**

Have the authors tried OTPF in such settings, or do they envision ways to extend it to support dense output tasks?

**Ethical Concerns:**

["NO or VERY MINOR ethics concerns only"]

**Final Justification:**

Thank you to the authors for providing a detailed rebuttal. The authors have effectively addressed my concerns, particularly with regard to OTPF’s adaptation to token compression, the handling of semantic complexity through QC-Comp, and the reduction of heuristics reliance with LP-Comp. Additionally, the VDC benchmark experiment successfully demonstrates OTPF's potential for dense-output tasks. Given these clarifications and improvements, I am satisfied with the responses and believe that the paper is now in an acceptable form. Therefore, I maintain my initial score and recommend acceptance.

**Limitations:**

yes

**Paper Formatting Concerns:**

No formatting error noticed.

**Quality:**

3

**Strengths And Weaknesses:**

Strengths:
1. OTPF achieves substantial token reduction (up to 90%) while preserving accuracy across multiple tasks, demonstrating strong practical value in resource-constrained settings.
2. The method is compatible with a wide range of backbones (ViT, CLIP, ResNet) and tasks (image classification, video understanding, multimodal QA), showing strong versatility.
3. OTPF can be used without any additional training, yet also supports end-to-end learning, making it easy to integrate into existing pipelines.
4. Outperforms or matches state-of-the-art baselines (e.g., uniform sampling, VideoMAE) in low-token regimes, validated across diverse benchmarks like ImageNet-1K and Kinetics-400.
Weaknesses:
1. OTPF’s effectiveness depends heavily on the quality of early-layer representations. If the backbone lacks strong discriminative power, selected tokens may be suboptimal.
2. OTPF compresses every input into a fixed number of tokens, regardless of the semantic richness or spatial/temporal complexity. This static compression may underrepresent informative regions in complex scenes and over-represent irrelevant parts in simple ones.
3. The token selection relies on hand-crafted heuristics (e.g., activation magnitude, channel max), which may not always correlate with semantic importance—especially in domain-shifted or adversarial settings.

---

> ### Author Rebuttal · Authors · 2025-07-31
>
> Thank you for your thoughtful and constructive review. We are pleased that you find our proposed method OTPF to be a lightweight, modular, and effective solution for efficient model deployment across diverse tasks. We appreciate your recognition of its versatility, strong performance under extreme token reduction, and ease of integration. Your insightful comments regarding early-layer representations, semantic complexity, and reliance on heuristics have helped us clarify and further articulate the motivations, design choices, and future directions of our work. We address your concerns and suggestions in detail below.
>
> ---
>
> ### 1. Early Layer Representations
>
> > OTPF’s effectiveness depends heavily on the quality of early-layer representations. If the backbone lacks strong discriminative power, selected tokens may be suboptimal.
>
> This is a very insightful observation, which is exactly our motivation for developing learnable and progressive compression, the heuristic-based methods in prior works. Our insight is that **OTPF's token compression relies less on the quality of early-layer representations** via optimizing the LLMs to conduct token compression and adapt to the early-layer representations.
>
> * Considering the heuristic-based approaches, *e.g.*, utilizing the correlation between text and image tokens or even uniform drop, they rely on the assumption that the early representations have to be sufficiently informative to mitigate the token drop. Instead, our learnable compression **jointly optimizes the LLM and early-layer representation** so that the LLMs can adapt to token compression.
> * More importantly, our compression is progressive and features a smooth reduction in the token number, which regulates the information loss happening at the ealy layers.
>
> ---
>
> ### 2. Token Reduction and Semantic Complexity
>
> > OTPF compresses every input into a fixed number of tokens, regardless of the semantic richness or spatial/temporal complexity. This static compression may underrepresent informative regions in complex scenes and over-represent irrelevant parts in simple ones.
>
> We agree that this is indeed a common confusion regarding token compression. Our design of XComp has attempted to address this problem from both aspects of inter-frame and intra-frame compression:
>
> * Regarding inter-frame compression, our method does not disregard semantic complexity. Instead, **our Question-Conditioned Compression (QC-Comp) is proposed to identify the informative frames** for emphasis and decrease the influence of irrelevant frames.
> * Regarding intra-frame compression, our learnable compression requires the LLMs to aggregate the information into a single token, which **supervises the LLMs to select and emphasize the informative regions into the final token**, otherwise, it cannot answer the questions correctly. We suggest that such a data-driven way is more adaptive than previous works' heuristics of localizing informative regions.
>
> As a side note, our paper emphasize **one token per frame** because of its landmark significance for token compression research in long videos and simplicity for implementation. We hope our work can inpire more adaptive and efficient token compression methods in future works.
>
> ---
> ### 3. Hand-crafted Heuristics in Token Selection
>
>
> > The token selection relies on hand-crafted heuristics (e.g., activation magnitude, channel max), which may not always correlate with semantic importance—especially in domain-shifted or adversarial settings.
>
> Thank you for this insightful question! The mismatch between heuristically selected tokens and desired information is a long-standing challenge for token compression in long videos, especially as **previous token compression entirely rely on heuristics**. Similar to the section above, we discuss from intra-frame and inter-frame token compression about our improvement in decreasing heuristics and directions for future work.
>
> * From an intra-frame level, our LP-Comp (learnable and progressive compression) is exactly proposed for LLMs to **learn the token compression via end-to-end optimization without relying on manually defined heuristics**.
> * From an inter-frame level, we acknowledge that our QC-Comp (question-conditioned frame compression) still relies on the heuristics of attention scores similar to prior works. However, the main novelty of QC-Comp is to enhance the efficiency of inter-frame token compression and *lost-in-the-middle* with our **segmented local attention**.
> * Connected with your insightful feedbacks, we envision an intriguing future work: extending our intra-frame learnable compression (LP-Comp) into the inter-frame compression QC-Comp to provide a heuristic-free token compression method. Despite its excitement, our submission did not attempt this as our LP-Comp was already a notable step towards heuristic-free compression, and extending the LLM to handle inter-frame compression would require significantly more computation resources for supervised fine-tuning. Therefore, we will gladly include these discussions into our future works.
>
> ---
>
> ### 4. Dense Output Tasks
>
> > Have the authors tried OTPF in such settings, or do they envision ways to extend it to support dense output tasks?
>
> Thank you for this constructive question! We hope to answer the question comprehensively to you.
>
> First, while OTPF primarily targets multiple-choice tasks on long videos, it is **fully capable of handling dense output tasks** such as video captioning. The primary reason we focused on multiple-choice tasks in the paper is due to the current landscape of long-video benchmarks, which predominantly evaluate models using multiple-choice formats. This is consistent with prior work in the domain of long video understanding, where the lack of dense-output benchmarks for long videos has been a limiting factor.
>
> To illustrate the capability of OTPF on dense-output settings, we conducted an additional experiment during the rebuttal phase on the VDC benchmark (proposed in the work Auroracap), specifically the detailed captioning subset, which requires dense textual descriptions. While this benchmark focuses on short video captioning, and is thus **beyond the scope of our main setting**, we believe it demonstrates that OTPF can generalize to dense generation tasks.
>
> As shown in the table below, our OTPF achieves nearly identical BLEU scores to VideoChat-Flash-2B:
>
>
> |  | BLEU@1 | BLEU@4 |
> | -------- | -------- | -------- |
> |  VideoChat-Flash-2B   | 7.1     | 1.6    |
> | OTPF  | 7.0   | 1.6    |
>
>
> As for the future direction, we believe the development of dense-output benchmarks tailored to long videos would be a valuable direction, and we appreciate your suggestion in motivating this line of research.

---

> > ### Comment · Reviewer_iYWa · 2025-08-05
> >
> > Thank you for the detailed rebuttal. The authors have effectively addressed my concerns, particularly regarding OTPF’s adaptation to token compression, the handling of semantic complexity through QC-Comp, and the reduction of heuristics reliance with LP-Comp. Additionally, the VDC benchmark experiment demonstrates OTPF's potential for dense-output tasks. Based on these clarifications, I am satisfied with the responses and maintain my initial score.

---

> > > ### Author Response · Authors · 2025-08-07
> > >
> > > Thank you for your continued engagement and thoughtful feedback. We’re pleased to hear that the VDC benchmark experiment was useful to you. Your input throughout this discussion has been invaluable, and we will incorporate all suggestions into the revised version. We appreciate your role in enhancing the quality of our work.

---

### Official Review · Reviewer_bQaD · 2025-07-03

**Clarity:** 3
**Significance:** 2
**Originality:** 3
**Rating:** 4
**Confidence:** 3

**Summary:**

This paper proposes a token compression framework termed XComp for long video understanding, addressing the challenge of token inefficiency in video-language models. The key idea lies in its hierarchical compression strategy: LP-Comp leverages LLM layers to learn progressive token compression (reducing frames to single tokens), improving density while avoiding heuristic failures of prior work; QC-Comp introduces query-conditioned frame selection via attention scores, mitigating positional bias through video segmentation. The combined approach enables extreme compression of 1 token per frame while preserving query-relevant information. Experiments demonstrate significant gains over VideoChat-Flash on long-video benchmarks.

**Questions:**

Please see weaknesses.

**Ethical Concerns:**

["NO or VERY MINOR ethics concerns only"]

**Final Justification:**

Generally speaking, the paper is well-motivated, thoughtfully designed, and demonstrates effective performance. However, one remaining limitation is that the effectiveness on fine-grained tasks in long videos has not been sufficiently verified. Even so, I am inclined to accept this paper and hope that the discussion on this limitation will be incorporated in the final version of the paper or appendix.

**Limitations:**

Yes.

**Quality:**

3

**Strengths And Weaknesses:**

- S1: The idea of letting LLM layers learn compression (LP-Comp) is well-motivated, moving beyond handcrafted heuristics.
- S2: QC-Comp’s attention-based segmentation elegantly handles position bias—a known limitation in long-context LLMs.
- S3: The paper is with promising empirical validation. Multi-benchmark improvements suggest generalizability.

- W1: While the proposed method could achieve comparable or better performance on several benchmarks, the efficiency comparison with baseline VideoChatFlash. I would like the authors to give more discussion and quantative results on time efficiency and FLOPs.
- W2: Compression works well on general videoQA. However, the performance on fine-grained video tasks like video OCR (like MME-VideoOCR) and reasoning benchmarks (like CLEVRER) should be considered to verifiy its generalization ability.

---

> ### Author Rebuttal · Authors · 2025-07-31
>
> Thank you for your thoughtful and encouraging review. We are glad that you find our proposed compression strategy—LP-Comp and QC-Comp—well-motivated and effective in addressing token inefficiency for long video understanding. We appreciate your recognition of our method’s empirical strength and generalizability across benchmarks.
>
> We are also grateful for your constructive suggestions regarding efficiency analysis and fine-grained benchmark evaluation. In the following, we provide quantitative comparisons on latency and FLOPs, and include new results on MME-VideoOCR and CLEVRER to assess generalization to short-video and fine-grained tasks. We believe these additions further clarify the strengths and limitations of our method.
>
> ---
>
> ### 1) Efficiency vs. VideoChat-Flash
>
> > While the proposed method could achieve comparable or better performance on several benchmarks, the efficiency comparison with baseline VideoChatFlash. I would like the authors to give more discussion and quantative results on time efficiency and FLOPs.
>
> Thanks for your suggestion to give more efficiency comparison! Here we would like to show the efficiency comparison on LVBench items (863 text tokens), measured over 10 runs on a single NVIDIA H200. Across 1k–4k frames, XComp reduces LLM TFLOPs by **~53–59%** and latency by **~45–58%**.
>
> | Frames | Model  | TFLOPs | Latency |
> |---:|---:|---:|---:|
> | 1,024 | VideoChat-Flash | 43 | 0.22 s |
> | 1,024 | XComp | 20 (**↓53%**) | 0.12 s (**↓45%**) |
> | 2,048 | VideoChat-Flash | 132 | 0.54 s |
> | 2,048 | XComp  | 58 (**↓56%**) | 0.25 s (**↓54%**) |
> | 4,096 | VideoChat-Flash | 448 | 1.56 s |
> | 4,096 | XComp | 187 (**↓58%**) | 0.66 s (**↓58%**) |
>
> ---
>
> ### 2) Short-video benchmarks (MME-VideoOCR & CLEVRER)
>
> > Compression works well on general videoQA. However, the performance on fine-grained video tasks like video OCR (like MME-VideoOCR) and reasoning benchmarks (like CLEVRER) should be considered to verifiy its generalization ability.
>
> We appreciate the suggestions from the reviewer on analyzing the performance of fine-grained perception, which is indeed a challenging scenario for video understanding. However, these two benchmarks mostly focus on **short videos** (MME-VideoOCR clips are mostly **≤10 s**, and CLEVRER videos are **5 s**), while **our XComp primarily targets the scope of efficiency and effectiveness on long videos (up to hours)**. However, we report the performance short-video results per the reviewer’s request.
>
> **MME-VideoOCR.** XComp and VideoChat-Flash-2B perform similarly  overall, where XComp shows stronger temporal grounding but slightly weaker spatial grounding.
>
> |  | Temporal Grounding | Spatial Grounding | Overall |
> |---|---:|---:|---:|
> | VideoChat-Flash-2B | 41 | **57** | **37.1** |
> | XComp | **45** | 52 | 35.4 |
>
> For subsets less tied to fine-grained spatial details, the two models are almost identical:
>
> |  | Attribute Recognition | Text-based Reasoning | Change Detection & Tracking |
> |---|---:|---:|---:|
> | VideoChat-Flash-2B | 56 | 42.6 | 42 |
> | XComp | 55.3 | 42.6 | 42 |
>
> **CLEVRER.**  Since CLEVRER training set is part of VideoChat-Flash pre-train data, both models perform strongly; XComp shows a small drop, as we only used ~2% of the original SFT data from VideoChat-Flash. Continued fine-tuning is expected to close this gap.
>
> | Model | Explanatory (Opt/Q) | Predictive (Opt/Q) | Counterfactual (Opt/Q) |
> |---|---:|---:|---:|
> | VideoChat-Flash-2B | 0.9463 / 0.8497 | 0.7789 / 0.5738 | 0.7775 / 0.4007 |
> | XComp | 0.9398 / 0.8406 | 0.7697 / 0.5600 | 0.7625 / 0.3652 |
>
> **Conclusions.** Although these short-video evaluations are out of our original scope, they reveal the discovery that our token compression only marginally impact the performance on fine-grained understanding. We appreciate the reviewer’s suggestion and will include these experiments and discussion to motivate future work on jointly optimizing long-video compression and fine-grained short-video perception.

---

> ### Comment · Reviewer_bQaD · 2025-08-06
>
> Thank you to the authors for providing additional experimental results.
>
> My concerns regarding Q1 have been well addressed. The latency results are impressive.
>
> For Q2, my original question was about fine-grained video tasks, rather than the short-video benchmark as mentioned in the authors’ response. Since understanding fine-grained text or subtle actions in long videos is more challenging, I am interested in exploring the boundary or limitations of the proposed compression methods. Specifically, I would suggest discussion on the performance of VideoOCR Bench on videos longer than 30 seconds. This would be more indicative of long-term understanding ability.

---

> > ### Author Response · Authors · 2025-08-07
> >
> > Thank you for your thoughtful follow-up and for emphasizing the distinction between short-video benchmarks and fine-grained understanding in longer videos.
> >
> > As requested, we conducted additional analysis on videos longer than 30 seconds from the MME-VideoOCR benchmark. The results are as follows:
> >
> >
> > | Model                                  | Video Tokens | MME-VideoOCR (Overall) | MME-VideoOCR (>30s) |
> > |----------------------------------------|--------------|----------------|--------------------|
> > | VideoChat-Flash-2B                     | 100%         | 37.1           | 49.1               |
> > | VideoChat-Flash-2B + continual-training*        | 100%         | 34.9           | 46.0               |
> > | XComp                                  | 50%          | 35.4           | 46.4               |
> >
> >
> > (* Continual training uses the same data as XComp; the same setting as Table D in the Appendix. It denotes using 2.5% of the VideoChat-Flash's training data for continual training, due to our limited computational resources. Note that this 2.5% is not identically distributed with VideoChat-Flash's original training data, as certain datasets are no longer available or could not be collected. This setting is designed to isolate the effect of training data versus our method and ensures a fair comparison.)
> >
> > These results confirm that **training data** has a pronounced effect on fine-grained tasks like VideoOCR, especially on longer videos. While our appendix shows minimal data impact on general QA, the effect becomes substantial on fine-grained benchmarks like MME-VideoOCR. This supports that the gap between XComp and VideoChat-Flash arises primarily from training data, not the method itself.
> >
> > Notably, XComp slightly outperforms the non-compressed baseline with the same training data, both overall and on the >30s subset, supporting the effectiveness of our compression method even under increased task difficulty.
> >
> > In this experiment added according to your request, we observe an interesting result: **why the ">30s" subset yields higher scores than the overall MME-VideoOCR**. This is because MME-VideoOCR's associated questions in long videos are often less challenging than those for shorter videos. For instance, in the most difficult category—Text_Recognition—only 4% of videos are longer than 30s, while 96% are short. As a result, the benchmark provides limited coverage of truly challenging, fine-grained tasks in long videos.
> >
> > In summary, while XComp performs well on long-video QA (as reported in the main paper), its advantage narrows on fine-grained tasks like OCR. We agree with the reviewer that understanding subtle content in long videos is critical. However, current benchmarks such as MME-VideoOCR offer limited coverage of temporally-distributed, fine-grained tasks. We believe richer benchmarks targeting these aspects would better expose the limitations—and strengths—of compression-based methods.

---

### Comment · Area_Chair_i3UF · 2025-08-03

Dear Reviewers,

Thanks for your hard work during the review process. We are now in the author-reviewer discussion period.

Please (1) carefully read all other reviews and the author responses; (2) start discussion with authors if you still have concerns as early as possible so that authors could have enough time to response; (3) acknowledge and update your final rating. Your engagement in the period is crucial for ACs to make the final recommendation.

Thanks,

AC

---

> ### Comment · Area_Chair_i3UF · 2025-08-05
>
> Dear Reviewers,
>
> As we're approaching the end of author-reviewer discussion period, please read the rebuttal and start discussion with the authors as soon as possible. If all your concerns have been addressed, please do tell them so. Please note that submitting mandatory acknowledgement without posting a single sentence to authors in discussions is not permitted. Please also note that __non-participating reviewers will receive possible penalties of this year's responsible reviewing initiative and future reviewing invitations.__
>
> Thanks,
>
> AC

---

### Note · Authors · 2025-08-16

We thank the reviewers, AC, and SAC for their time, constructive feedback, and thoughtful engagement throughout the process. Our paper initially received positive scores from the reviewers, and we appreciate the opportunity to clarify our contributions and address raised concerns.

Our work introduces XComp, a hierarchical token compression framework for long video understanding, combining learnable progressive compression (LP-Comp) and query-conditioned compression (QC-Comp). This design reduces tokens to 1/frame while aiming to preserve query-relevant information and mitigate positional bias. Experiments show improvements over VideoChat-Flash on multiple long-video benchmarks, and the approach also applies to other backbones, as demonstrated on LLaVA-Next-Video.

In response to reviewer requests, we added:
- Efficiency – On LVBench items, XComp reduced LLM TFLOPs by ~53–59% and latency by ~45–58% vs. VideoChat-Flash.
- Fine-grained evaluation – On MME-VideoOCR (>30s) and CLEVRER, performance gaps with baseline appear mainly due to training data; XComp matched or slightly exceeded baseline when data was controlled.
- Generalization – Gains on LLaVA-Next-Video with fewer tokens/frame.
- Ablations – Completed on LVBench and LongVideoBench, supporting the contributions of LP-Comp, QC-Comp, and design choices.
- Dense-output test – Comparable results to baseline on VDC captioning benchmark.

Reviewers iYWa and Ur6N stated their concerns were addressed and increased or maintained their positive scores. Reviewer bQaD confirmed Q1 was fully addressed (“latency results are impressive”). We thank all reviewers for their feedback, which has helped improve the scope and clarity of our work, and we will incorporate their suggestions into the final version.

---

### Decision · Program_Chairs · 2025-09-17

**Decision:**

Accept (poster)

**Comment:**

This paper presents a hierarchical token compression framework for long video understanding to address the token inefficiency issue in video-language models. Reviewers acknowledged the contribution and strong performance of the proposed method, while initially raising concerns regarding its generalization ability and missing important experiments.

After the rebuttal, the authors addressed most of the concerns, and all reviewers agreed to accept this paper. AC read all the reviews, author rebuttals, and the paper, and believes this is a strong paper and recommends acceptance.